# Construction of the Cellulose Nanofibers (CNFs) Aerogel Loading TiO_2_ NPs and Its Application in Disposal of Organic Pollutants

**DOI:** 10.3390/polym13111841

**Published:** 2021-06-02

**Authors:** Kang Li, Xuejie Zhang, Yan Qin, Ying Li

**Affiliations:** Key Laboratory of Colloid and Interface Chemistry of State Education Ministry, Shandong University, Jinan 250100, China; 202012265@mail.sdu.edu.cn (K.L.); xuejzhang@foxmail.com (X.Z.); qy18366104018@yahoo.com (Y.Q.)

**Keywords:** cellulose aerogel, chemical cross-linking, photocatalysis

## Abstract

Aerogels have been widely used in the adsorption of pollutants because of their large specific surface area. As an environmentally friendly natural polysaccharide, cellulose is a good candidate for the preparation of aerogels due to its wide sources and abundant polar groups. In this paper, an approach to construct cellulose nanofibers aerogels with both the good mechanical property and the high pollutants adsorption capability through chemical crosslinking was explored. On this basis, TiO_2_ nanoparticles were loaded on the aerogel through the sol-gel method followed by the hydrothermal method, thereby the enriched pollutants in the aerogel could be degraded synchronously. The chemical cross-linker not only helps build the three-dimensional network structure of aerogels, but also provides loading sites for TiO_2_. The degradation efficiency of pollutants by the TiO_2_@CNF Aerogel can reach more than 90% after 4 h, and the efficiency is still more than 70% after five cycles. The prepared TiO_2_@CNF Aerogels have high potential in the field of environmental management, because of the high efficiency of treating organic pollutes and the sustainability of the materials. The work also provides a choice for the functional utilization of cellulose, offering a valuable method to utilize the large amount of cellulose in nature.

## 1. Introduction

With the rapid development of industry and agriculture, the problem of environmental pollution is getting worse. Organic pollution which can cause serious damage to biological and ecological systems has caught more and more attention all over the world [1]. Adsorption [2,3,4,5,6,7,8], membrane separation [9], and the REDOX method [10] have been gradually developed for the treatment of organic pollutants. Adsorption has become a widely used method due to its advantages, including simple operation, high removal efficiency, and less secondary pollution risk. Aerogels have been recognized as good adsorbents because of their porous structure and large specific surface area [2].

Natural polymers, such as chitosan and cellulose, have been commonly used in the construction of aerogels [11,12,13]. Nanocellulose, such as cellulose nanofibers (CNFs), has a high potential to prepare aerogels due to their widest source, sustainability, biodegradability, and plentiful polar groups [14,15,16]. However, because CNF aerogels mainly rely on hydrogen bond crosslinking, the poor mechanical stability, especially in a water-presenting environment [15] is the main bottleneck limiting its practical application. To solve this problem, chemical crosslinking between cellulose chains can be applied to enhance its stability. Carboxyl groups, aldehyde groups, etc. have been reported as being introduced into cellulose chains to obtain chemically cross-linked cellulose aerogels that are stable in water [15,17].

Nevertheless, the enrichment of organic pollutants in the hydrogel or aerogel does not completely solve the pollution problem. The further treatments are still needed to degrade the pollutants. Heterogeneous photocatalytic degradation has been recognized as one of the most efficient methods to eliminate the organic pollutants. The photocatalyst can generate photogenerated electrons (e^-^) and holes (h^+^) under the excitation of ultraviolet light or visible light. The e^-^ and h^+^ act on the surrounding O_2_ and H_2_O to convert them into active oxygen species, such as •OH and •O^2−^. Electrons, holes, and active oxygen species can degrade organic pollutants into CO_2_ and H_2_O, and then solve the environmental pollution problems caused by these organic pollutants [18,19,20]. TiO_2_ nanoparticle (NP) is an excellent type of candidates used as photocatalyst in decontamination of organic compounds, due to the excellent chemical stability, non-toxicity, and excellent photocatalytic performance [21,22]. It has been indicated that TiO_2_ can efficiently degrade almost all organic pollutants into CO_2_, H_2_O, and other simple inorganic substances under UV light [23]. While in practical applications, the NPs need carriers to ensure their dispersion and activity. If the photocatalyst could be loaded on cellulose aerogel, on one hand the cellulose network could be used as the template of TiO_2_ NPs, on the other hand, the adsorption of pollutants by cellulose aerogel and the degradation of pollutants by photocatalyst could be realized synergistically.

Based on the above, in this paper, 3,3-dithiodipropionic acid dihydrazide (DAD) molecules were chosen as the crosslinking agent, for the purpose of constructing the chemically cross-linked CNF Aerogels with good mechanical properties. The other prospect for this choice is that the hydrazide and disulfide groups inducted by the crosslinker might enhance the adsorption of the Ti precursor, thereby the TiO_2_ nanoparticles could be loaded on the CNF Aerogel firmly through the sol-gel and hydrothermal processes. It was found that such prepared composite aerogel has mesoporous structure, large surface area and high capability for adsorbing organic molecules. The measurement of the performance in degradation of pollutants carried out afterwards verified the high potential of the prepared TiO_2_@CNF Aerogels in the treatment of organic pollutants.

## 2. Materials and Methods

### 2.1. Materials

To ensure the material repeatability, ashless filter paper was used in this work to produce CNF, which was purchased from Aoke Filter Paper Factory, Taizhou, China. 2,2,6,6-tetramethylpiperidine-1-oxyl radical (TEMPO), Dimethyl 3,3’-dithiodipropionate, Hydrazine, *N*-hydroxysuccinimide (NHS), and *N*-ethyl-*N*-(3-(dimethylamino)propyl)-carbodiimide (EDC) were all purchased from McLean Reagent Co., Ltd. (Shanghai, China); Sulfuric acid was purchased from Laiyang Kangde Chemical Co., Ltd. (Shandong, China); Ethanol, methanol, and glacial acetic acid were purchased from Tianjin Fuyu Fine Chemical Co., Ltd. (Tianjin, China); Other reagents were purchased from Sinopharm Chemical Reagent Co., Ltd. (Shanghai, China); ULUPURE ultrapure water machine was used to provide ultrapure water for all experiments.

### 2.2. Preparation of CNF Suspension and Oxidized Cellulose Nanofibers (CNFs-COONa)

The CNF suspension with enhanced mechanical properties was obtained from the acid hydrolysis of filter papers using 64 wt.% sulfuric acid at 45 °C for 45 min to remove the amorphous region [24].

The CNF suspension was oxidized to CNFs-COONa by using TEMPO as the catalyst [25]. TEMPO (0.06 g) and NaBr (1.0 g) were added into 1 wt.% CNF suspension (500 mL), then 30 mL NaClO solution was added drop-by-drop, the pH was controlled at about 10 by NaOH solution (0.1 mol/L) simultaneously. The suspension was then centrifuged and dialyzed until the pH of CNFs-COONa suspension did not change. CNFs-COONa suspension was then sonicated to make it evenly dispersed. The carboxyl content of CNFs-COONa was 1.15 ± 0.04 mmol/g measured by conductometric titration.

### 2.3. Preparation of Oxidized Cellulose Nanofibers (CNFs-CHO)

Aldehyde-modified CNFs (CNFs-CHO) were prepared by oxidizing the CNF suspension with NaIO_4_. NaIO_4_ (1.5 g) was dissolved in 1 wt.% CNF aqueous suspension (200 mL) [26], the mixture was stirred for 2 h in the dark. The product was dialyzed to neuter, and CNFs-CHO suspension with different solids’ content could be obtained through filtration or dilution of the 1 wt.% CNFs-CHO mixture.

### 2.4. Hydrazide-Modified CNF Synthesis (CNFs-DAD)

3,3-Dithiodipropionic acid dihydrazide (DAD) was prepared according to the reference (yield 55~75%) [27]. CNFs-DAD was prepared by amidation reaction, which occurred between the carboxyl groups on the CNFs-COONa and the amino groups on DAD. The detailed synthetic procedure was as follows. EDC·HCl was added to the system as the dehydrator, and NHS was added to activate the leaving group. The mixture was stirred at room temperature for 24 h in the dark, and then dialyzed. The product was placed in a 4 °C environment.

### 2.5. The Preparation of Chemically Crosslinked CNF Aerogel

The chemically crosslinked CNF hydrogel was prepared by mixing the CNFs-DAD suspension and CNFs-CHO suspension with the same concentration (0.5, 1.0, 2.0, or 3 wt.%). The chemical crosslinking was formed by the reactions between the aldehyde groups on the CNFs-CHO and the amino groups of DAD. The mixture was stirred for 5 to 10 min and then gelling on standing. The reactions in the synthesis route of chemically crosslinked CNF Aerogel were shown in Appendix A. Then the solvent of the prepared cellulose hydrogels was replaced with ethanol, followed by freeze-drying for 48 h to obtain the CNF Aerogel.

### 2.6. Synthesis of TiO_2_@CNF Aerogel

TiO_2_@CNF Aerogel was prepared by a combination of the sol-gel method and the hydrothermal method. The prepared CNF Aerogel was soaked into the mixed solution of Ti(OC_4_H_9_)_4_ and absolute ethanol (V:V = 1:4) for 30 min. Then the aerogel was taken out and those Ti(OC_4_H_9_)_4_ not adsorbed were washed out by absolute ethanol. Afterwards, the gel was immersed into the mixed solution of absolute ethanol and water in a volume ratio of 4:1 and then pH was adjusted to 3.0 with diluted hydrochloric acid. Then, the gel was kept for 2 h to make sure the adsorbed Ti(OC_4_H_9_)_4_ was completely hydrolyzed. The intermediate was then placed into a PTFE-lined hydrothermal reactor, reacted at 120 °C for different durations (2, 4, and 6 h) to determine the optimum reaction time. Freeze-drying was then carried out to obtain TiO_2_@CNF Aerogel. The above absorption, hydrolysis, and hydrothermal processes were repeated 1–5 times to obtain the aerogel loading with different amounts of TiO_2_ NPs.

### 2.7. Characterization Methods

X-ray diffraction (XRD) determination was performed on a Rigaku Dmax-rc X-ray diffractometer in the angular range of 2θ = 10°–80°. The scanning electron microscope (SEM) (Hitachi SU8010, Tokyo, Japan) image was taken on JEOLLTD JSM-6700F, while the transmission electron microscope (TEM) (JEOL JEM, 1011, Tokyo, Japan) that was used was the HT-7700. Fourier transform infrared (FT-IR) spectra were recorded on the ATR-FTIR spectrometer (Thermo Fisher, Nicolet iS5, Waltham, MA, USA) in the 650–4000 cm^−1^ region. The compression performance of the aerogel was measured by a texture analyzer (TA, TMS-PRO, FTC, USA). The surface area was identified by the N_2_ adsorption–desorption isotherms at 77 K based on the BET model (Micromeritics, USA). X-ray photoelectron spectroscopy (XPS, Thermo Fisher, Thermon ESCALAB 250XI spectrometer, USA) were recorded to characterize the elemental states.

### 2.8. Adsorption and Degradation of Rhodamine B (RhB)

TiO_2_ can degrade a variety of organic pollutants and dyes into CO_2_ and H_2_O [24]. In this work, Rhodamine B was selected as the sample dye. The prepared aerogels were cut into thin membranes to promote the contact between the UV light and TiO_2_ NPs embedded in the aerogels. Then the thin TiO_2_@CNF Aerogel was put in the RhB solution in the dark (10~200 mg/L) until reaching adsorption saturation. Then the photocatalysis was started by exposing the adsorption saturated aerogel to UV light with constant stirring. Ultraviolet light is provided by a UV lamp, which was 15 cm from the surface of liquid. The concentration of RhB was examined by a UV spectrophotometer (Shimadzu, Japan) at λ = 554 nm, which is the maximum adsorption wavelength of RhB [1], and the removal rate was characterized by the following formula:Removal rate = (*C*_0_ − *C*_T_)/*C*_0_ × 100%(1)
where *C*_0_ was initial concentration and *C*_T_ was concentration at time T.

## 3. Results and Discussion

### 3.1. Characterization of the Chemically Crosslinked CNF Aerogel

The FTIR spectra of the different kinds of modified CNFs were shown in Figure 1. The peaks at 3320, 2913, 1422, 1155, 1111, and 1050 cm^−1^ were all typical peaks of cellulose [28]. It could be found that, for the cellulose sample treated by different ways (CNFs, CNFs-CHO, CNFs-COONa, and CNFs-DAD), no obvious change could be observed in the positions of the peaks, which means that the basic chemical structure of cellulose was retained. Aldehyde modified CNFs (CHO−CNFs) had an obvious absorption peak at 1740 cm^−1^, indicating that carbonyl groups (C=O) appeared. With the addition of saddle-shaped double peaks at 2750 and 2850 cm^−1^ which indicates the Fermi resonance of the aldehyde groups, the introduction of aldehyde groups was certified. The peak at 1740 cm^−1^ was assigned to the stretching vibration of carbonyl groups (C=O) in CNFs-COONa, while the peak at 1660 cm^−1^ was assigned to the stretching vibration of carbonyl groups (C=O) in CNFs-DAD, which was varied because of the electronic effect. The sharp double peaks at 3300 cm^−1^ indicated the presence of primary amine groups, which was consistent with the structure of CNFs-DAD. The results of the conductometric titration also proved the chemical functionalization of CNFs-DAD and CHO-CNFs, as shown in Appendix A.

After simple mechanical mixing, the reaction between hydrazide groups on CNFs-DAD and the aldehyde groups on CHO-CNFs occurred during the mixing and hydrazone group formed between them. The chemical crosslinking and the hydrogen bond between the CNFs provided a stable network structure and formed a firm hydrogel eventually. After solvent exchange and freeze-drying, chemically crosslinked CNF Aerogel was successfully prepared. The preparation of the hydrogel was shown schematically in Figure 2.

The pore structure of the aerogel was tested with the sample prepared by 2 wt.% CNFs-CHO suspension and 2 wt.% CNFs-DAD suspension. The microstructure of the prepared CNF Aerogel was observed by SEM. As what was shown in Figure 3a,b, the internal structure of aerogel appeared as a hierarchically porous structure, and had some mesopores among cellulose fibers (Figure 3b) which can increase the interfacial area, while macropores are conducive to the diffusion of substances in aerogels, both can enhance the adsorption capacity.

The N_2_ adsorption–desorption isotherm was shown in Figure 3c, which showed a type IV nitrogen adsorption–desorption hysteresis isotherm, and the specific surface area was 123.1 m^2^/g for the aerogel. As seen in Figure 3d, the peak volume of the aperture was at 16 nm, which corresponds to the mesoporous size.

The mechanical properties of CNF Aerogels were determined by TA. The stress–strain curve of water-saturated CNF Aerogel was shown in Figure 4a. When the compressive strain was lower than 60%, compressive stress showed a platform and the rising trend was slow. When compressive strain exceeded 70%, compressive stress began to rise rapidly. At low compressive strain, the deformation of the aerogel was mainly due to the bending and collapse of macropores, while as the compressive strain rose above 70%, chemical crosslinking may be destroyed. This also indicates that the higher the cellulose content in the aerogel, the stronger the mechanical properties of the aerogel, which is caused by more crosslinking sites.

Figure 4b showed the shape recovery of chemically crosslinked CNF Aerogel that was prepared by the 2.0 wt.% CHO-CNF suspension and 2 wt.% CNFs-DAD suspension. In the air, the shape recovery percentage significantly declined with the rise of compressive strain. When the strain exceeded 80%, CNF Aerogel was difficult to restore the original shape, which was consistent with the result of the stress–strain curve. However, the CNF Aerogel maintained excellent mechanical properties in water (Figure 4c–e). When the compressive strain reached up to 80%, the shape recovery percentage can still attain 90%. This suggested that hydrone can act as a shape recovery agent to maintain the original shape of the aerogel, which is beneficial for the CNF Aerogel used in water.

### 3.2. Characterization of the TiO_2_@CNF Aerogels

XRD measurements were taken to characterize the crystal structure of the TiO_2_ nanoparticles formed inside the CNF Aerogel.

As shown in Figure 5a, the black line indicated unloaded TiO_2_ prepared by the sol-gel method (not hydrothermally treated), and the lines of other colors indicated that TiO_2_ prepared by the sol-gel method followed by a hydrothermal reaction with different times. When hydrothermal treatment was carried out for 1 h, almost no diffraction peak appears, indicating that no crystal phase formed in that short time. When the hydrothermal time was 2 h, the intensity of diffraction peak was relatively weak and the peak was wide, indicating that the crystallinity was poor. Continuing to increase the hydrothermal time clearly determined that the intensity of diffraction peak gradually became stronger. When hydrothermal time reached up to 4 h, it can be clearly seen that the anatase TiO_2_ diffraction peaks (2θ = 25.3°, 37.6°, 47.8°, 54.4°, 62.7°, 68.8°, 70.3°, and 75.1°, respectively) [29] were high. When the hydrothermal time was further increased, the shape of the peak did not change significantly, indicating that the further prolongation of the hydrothermal time does not have obvious influence on the crystallinity of TiO_2_. This indicates that a transition of TiO_2_ particles from amorphous to crystalline occurred in the hydrothermal process, and the optimal duration of the hydrothermal treatment is 4 h.

The influence of the hydrothermal reaction on the structure of the CNFs in the aerogel was also tested. Figure 5b showed the characteristic peak of CNF Aerogel and the peak values at 2θ = 14.6°, 16.3°, and 22.6°, which was consistent with cellulose Ⅰ [28]. Thus, the basic crystal structure of cellulose was not destroyed in the hydrothermal process.

The crystal structure of TiO_2_ loaded on CNF Aerogel was tested to get the optimum hydrothermal reaction time. In Figure 5c, we can see that in the case of carrying out the hydrothermal reaction for 2h, anatase TiO_2_ appeared in the composite, and the peak became sharper when the hydrothermal reaction time increased to four hours. When reaction time was more than 4 h, the change of crystal structure was not obvious, which was consistent with the result given by Figure 4a. In conclusion, in terms of minimum energy and time, the optimal reaction condition for the conversion of TiO_2_ from amorphous to anatase was conducting hydrothermal reaction for 4 h at 120 °C.

It can be seen in Figure 6a that the TiO_2_@CNF Aerogel retained the original porous structure, while the size of the macropore decreased from 50 to 20 μm. This was mainly because the load of TiO_2_ NPs occupies the surface of the macropores and reduces the size of the macropores. Figure 6b showed the morphology of TiO_2_ on the cross-section of CNF Aerogel, in which spherical TiO_2_ nanoparticles were uniformly dispersed with the diameter of about 50 nm.

The nitrogen adsorption–desorption hysteresis isotherm of TiO_2_@CNF Aerogel showed a type IV curve (Figure 6c), the same as that of the CNF Aerogel. However, when the TiO_2_ nanopaticles were loaded on the CNF Aerogel, the specific surface area increased from 123.1 to 330 m^2^/g (Figure 6c), which was due to the loading of TiO_2_ NPs. The peak of the aperture of TiO_2_@CNF Aerogel was around 35 nm after TiO_2_ NPs were loaded on the surface of aerogel, which was larger than that unloaded (Figure 6d). This may be because that partial chemical crosslinking was destroyed by the adhered TiO_2_ NPs so the average size of pores increased.

The XPS spectrum of TiO_2_@CNF Aerogel revealed that the composite contains five elements (Ti, O, C, S, and N), in which the chemical binding energies for Ti 2p, O 1s, and C 1s were 458.4, 529.7, and 286.4 eV, respectively, as shown in Figure 7a. Higher-resolution XPS spectra were tested to further understand the chemical status of elements in the TiO_2_@CNF Aerogel. The XPS spectra of C 1s was shown in Figure 7b and five peaks were observed after multi-peak Gaussian fitting. The peak at 284.6 eV was ascribed to carbon atoms in C–C bonds. The peak at 286.26 eV was attributed to the C–O bond and the peak at 287.26 eV corresponded to C–O–C bond. The peak at 288.15 eV matched with C–O–Ti bond [30,31]. The presence of the C–O–Ti structure reveals that the chemical bond was formed between TiO_2_ NPs and CNFs, which made the TiO_2_ NPs adsorbed on the CNFs. Figure 7c displayed the O 1s XPS spectra. The O 1s peak at 529.88 eV was corresponding to lattice oxygen of Ti-O, while the higher binding energy of 531 eV was assigned to hydroxyl groups (O–H), and the peak at 532.7 eV was attributed to the O–C bond. This matched the structure of cellulose and TiO_2_. Figure 7d presented the Ti 2p spectra. The binding energy of Ti 2p3/2 and Ti 2p1/2 were 458.6 and 464.4 eV, respectively, which were close to those of pure anatase phase (i.e., 457.9 and 463.8 eV). This suggests that most of the Ti was still in the form of anatase phase, with only a small amount of bonds.

According to the above results, it could be concluded that the prospected assembly process of TiO_2_ NPs in the CNF Aerogel occurred, as is schematically illustrated in Figure 8a. In Step 1, as the precursor of TiO_2_, Ti(OC_4_H_9_)_4_ was adsorbed on the surface of the CNF Aerogel, especially on the crosslinkers (Figure 8b). Subsequently, the TiO_2_ crystal nucleus formed and grew, then TiO_2_ NPs with good crystallinity were successfully loaded on the CNF Aerogel. (Figure 8c). When the above steps were repeated four more times, the amount of the TiO_2_ NPs increased.

### 3.3. Application of CNF Aerogel and TiO_2_@CNF Aerogel—Adsorption and Degradation of RhB

#### 3.3.1. Adsorption of RhB by CNF Aerogel

Figure 9a shows the adsorption curve of chemically crosslinked CNF Aerogel for different concentrations of RhB, in which the ordinate represents the amount of RhB adsorbed by aerogel per unit mass. It could be found that the adsorption amount gets higher rapidly in the first 30 min, and reached saturation after about 60 min. In the early stage of adsorption, the adsorption rate increased with the increase of concentration (Figure 9a). In order to ensure the adsorption equilibrium, the adsorption time was extended to 120 min to test the pollutant removal efficiency. As shown in Figure 9b, the adsorption efficiency decreased with the increase of RhB concentration. The optimum adsorption efficiency was about 58% when the concentration was 10 mg/L, while when the concentration of RhB increased to 200 mg/L, the removal efficiency was only about 17%.

#### 3.3.2. Degradation of RhB by TiO_2_@CNF Aerogel

Figure 10a showed the degradation rate of TiO_2_@CNF Aerogel with different loading times of TiO_2_. The concentration of RhB in the experiment was 200 mg/L and degradation experiments were carried out under ultraviolet lamps. When the loading times increased from 1 to 3, the degradation rate and efficiency of RhB increased gradually. However, when the loading time is up to 4, the degradation efficiency decreased slightly. This may be because the excessive TiO_2_ NPs covered the adsorption site of CNF Aerogel, then the adsorption of RhB was limited (Figure 10b). Therefore, to balance the adsorption and degradation, we chose three times of TiO_2_ loading as the best reaction condition to carry out photocatalytic degradation experiments on different concentrations of RhB (10~200 mg/L).

In Figure 11a, we can see that RhB can be absorbed without illumination, so the concentration of RhB dropped slightly when no light was applied. After exposure to ultraviolet light, RhB began to be degraded rapidly. Within 120 min, RhB had a degradation efficiency of up to 99% when the concentration of RhB was 10 mg/L. With increases of dye concentration, the degradation efficiency gradually decreased. When the concentrations of RhB were 50, 100, and 200 mg/L, their degradation efficiencies were 90.5%, 82.9%, and 72.1%, respectively, within 120 min. When the illumination time exceeded 240 min, the degradation efficiency of all samples could exceed 90%. This indicates that the TiO_2_@CNF Aerogel had an excellent pollutant treatment efficiency after loading TiO_2_.

The reusability of TiO_2_@CNF Aerogel is shown in Figure 11b. It can be observed that the degradation efficiency of the TiO_2_@CNF Aerogel decreased slightly with the increase of using times. However, after five cycles, the degradation efficiency was still 72% of that of the first use. This indicates that the TiO_2_@CNF Aerogel composite material had fairly good stability. Due to the good adhesion of TiO_2_ nanoparticles on the surface of CNF Aerogel, the synergistic effect of the adsorption and the photocatalysis was ensured. Therefore, the composite material has a great recycling performance.

The performance of the TiO_2_@CNF Aerogel produced in this work in the adsorption and degradation of pollutants were compared with other cellulose-based materials reported in the literature, as listed in Table 1. We can see that the TiO_2_@CNF Aerogel in this work has good performance in both adsorbing and degrading pollutants.

Depending on the nature of the prepared material, the environmental friendliness, convenience, and high efficiency could be the most valuable advantages of the TiO_2_@CNF Aerogel, which would highly benefit its application in the field of water pollution treatment. This method can thoroughly treat the pollutants in water without causing secondary pollution. As shown in Figure 12, the prepared aerogels could be cut into thin membranes and float on the surface of water, which can also be moved up and down mechanically, so that the aerogel can adsorb pollutants under water and degrade the pollutants on the surface of water under sunlight. After the water was purified, the recycling of the TiO_2_@CNF Aerogels can be easily realized.

## 4. Conclusions

Cellulose aerogel with good mechanical stability was prepared by chemical modification and cross-linking of CNFs, and titanium dioxide nanoparticles were loaded inside the CNF Aerogel by the combination of the sol-gel method and the hydrothermal method. The preparation of the composite material takes filter paper as the main raw material, and adds a small amount of the chemical reagents for modification, cross-linking, and functionalization. In the preparation process, DAD not only serves as a crosslinking agent to help build three-dimensional network structure, but also provides a large number of sites for the loading of TiO_2_ precursor, which is conducive to the stability of the material during recycling. The prepared composite aerogel has both good adsorption capacity for organic molecules and photocatalytic capability, which ensured the efficient adsorption firstly and the complete degradation of organic pollutants afterwards. The composite aerogel material has a low density, which makes it very convenient to be collected and recycled. After recycling, due to the degradability of the cellulose matrix itself, the aerogel can be degraded naturally.

The presented TiO_2_@CNF Aerogels provide a choice for the functional utilization of cellulose in the field of environmental management. This combination of adsorption and photocatalysis proposes an effective solution for the treatment of a large number of organic pollutants in water. The research result found in this paper using filter paper as the material also reveals the possibility utilizing other sources of cellulose for this prospect, offering a recycling method for a large amount of cellulose-based waste materials in nature, such as waste paper and coconut shells, which highly benefits the sustainable development of society. Therefore, the utilization of a large number of natural polymer resources and non-toxic photocatalysts to develop environmentally friendly composite gel materials is not only conducive to the effective utilization of natural resources, but also provides a sustainable way to solve the problem of environmental pollution, which should be a major development direction in this field. Moreover, further expanding the absorption of visible light by photocatalysts and making better use of sunlight is also one of the meaningful development directions in this field.

## Figures and Tables

**Figure 1 polymers-13-01841-f001:**
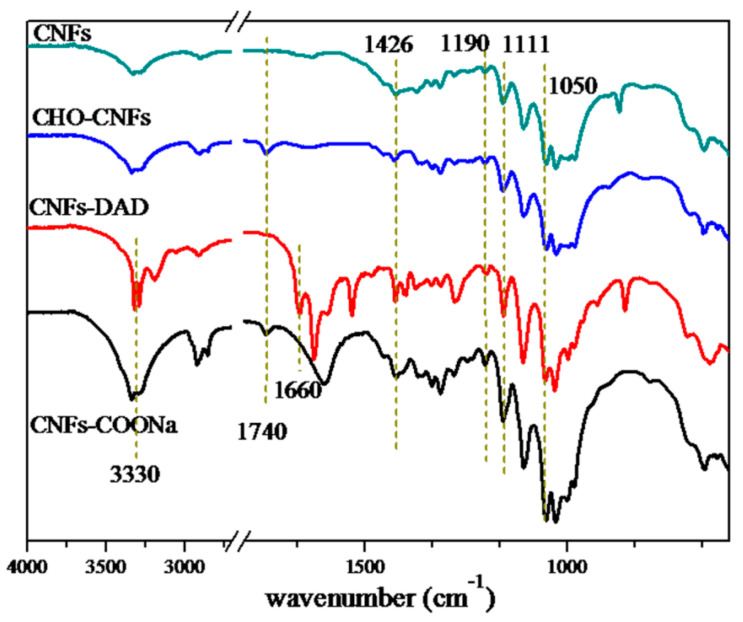
FTIR spectra of CNFs, CNFs-CHO, CNFs-DAD, and CNFs-COONa.

**Figure 2 polymers-13-01841-f002:**
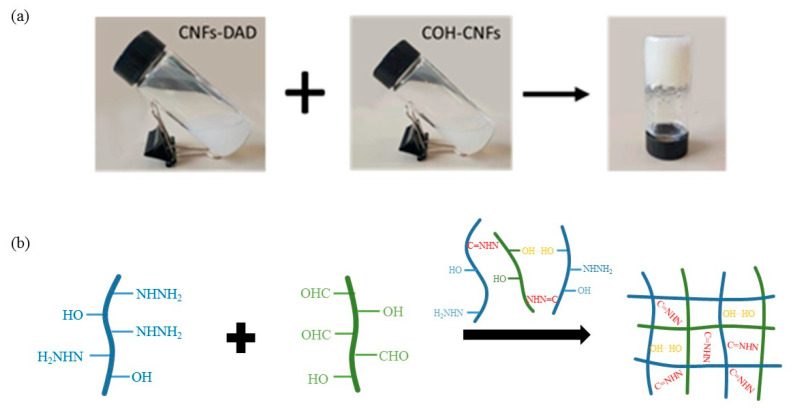
Schematic representation of chemically cross-linked CNF hydrogel and the reactions in the crosslinking process: (**a**) Photos of the gelation, (**b**) Schematic diagram of chemical reactions between chains in the cross-linking process.

**Figure 3 polymers-13-01841-f003:**
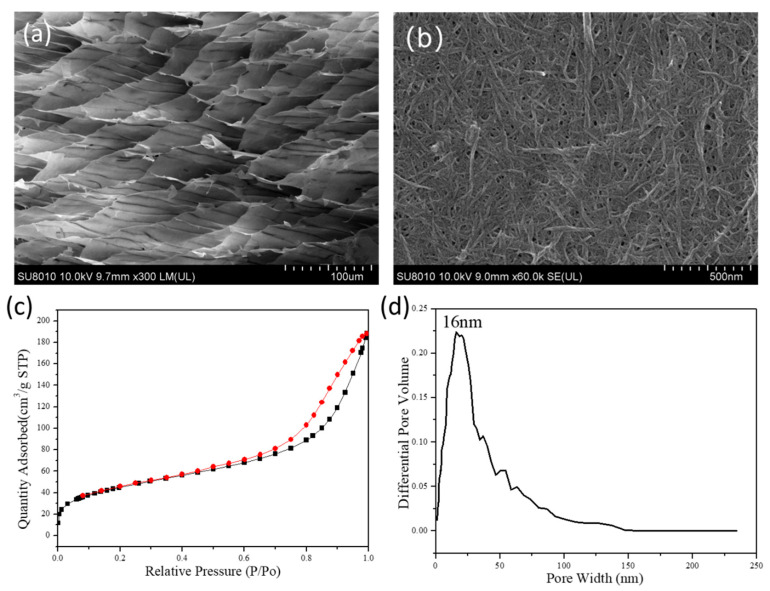
SEM images of the CNF Aerogel with progressively increased magnification: (**a**) honey comb porous structure, (**b**) mesoporous structure on the cell walls, (**c**) BET N_2_ adsorption–desorption isotherm, and (**d**) pore size distribution of CNF Aerogel.

**Figure 4 polymers-13-01841-f004:**
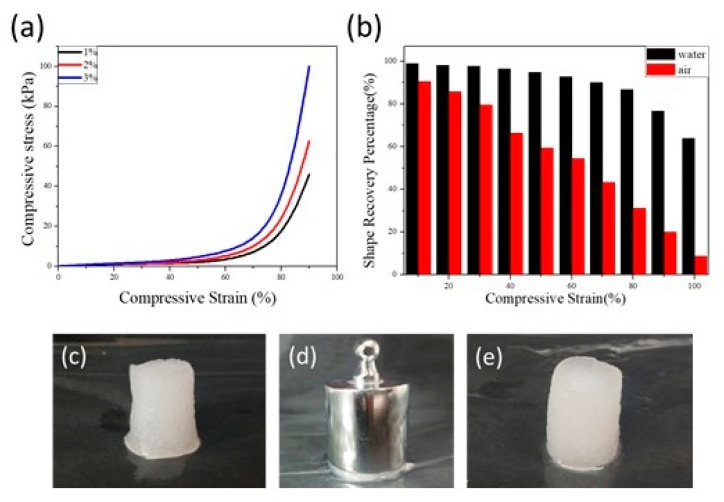
(**a**) Compressive stress–strain curves of water-saturated CNF Aerogels prepared, (**b**) Shape recovery percentage of CNF Aerogel under different compressive strains (0~95%) after 10 cyclic compressions both in water and air, (**c**–**e**) Photograph of manually compressed water-saturated CNF Aerogel.

**Figure 5 polymers-13-01841-f005:**
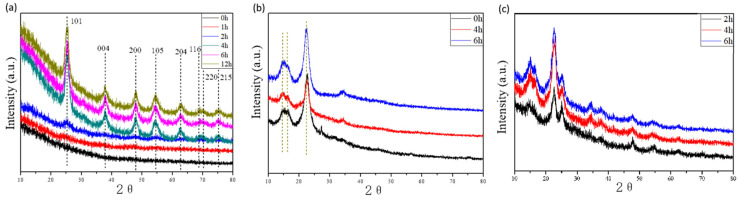
XRD patterns of: (**a**) TiO_2_ prepared by sol-gel method with different hydrothermal time, (**b**) chemically crosslinked CNF Aerogel with different hydrothermal time, and (**c**) TiO_2_@CNF Aerogel hydrothermal treated with different time.

**Figure 6 polymers-13-01841-f006:**
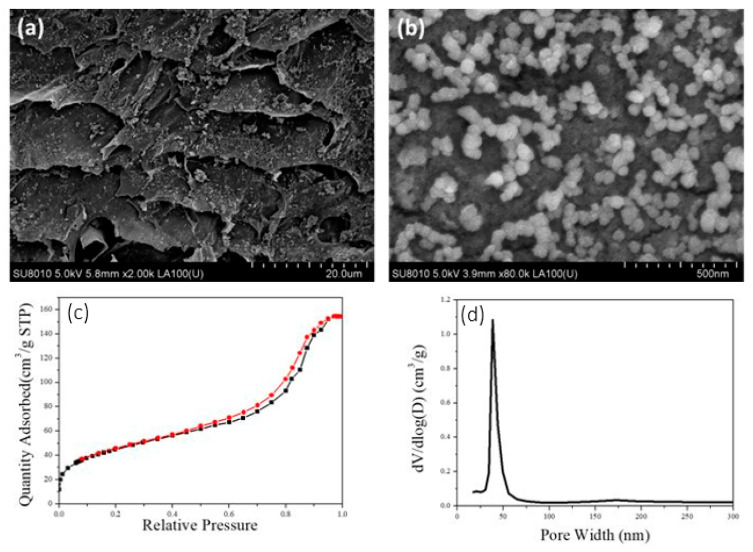
(**a**) and (**b**) The SEM image of TiO_2_@CNF Aerogel in different resolution, (**c**) N_2_ adsorption–desorption isotherm, and (**d**) pore size distribution of TiO_2_@CNF Aerogel.

**Figure 7 polymers-13-01841-f007:**
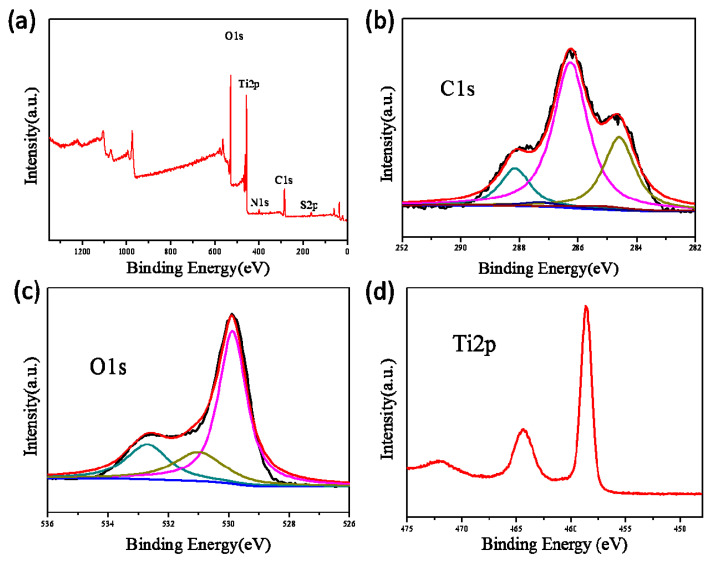
(**a**) Survey XPS spectra of TiO_2_@CNF Aerogel, (**b**) C 1s, (**c**) O 1s, and (**d**) Ti 2p XPS spectra of TiO_2_@CNF Aerogels, respectively.

**Figure 8 polymers-13-01841-f008:**
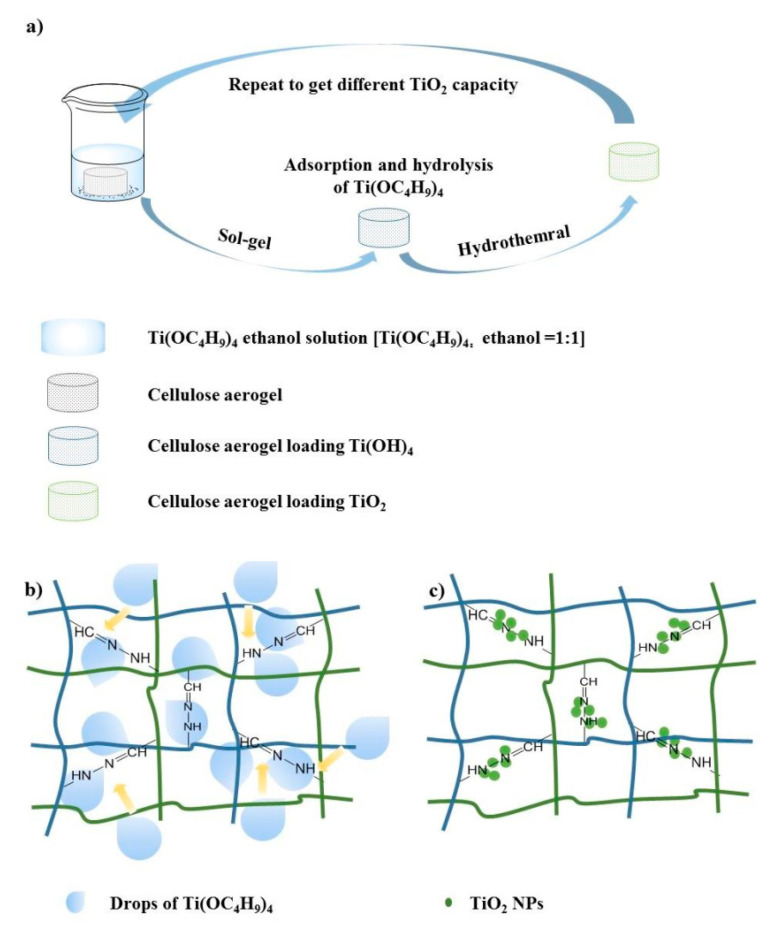
Schematic illustration of the TiO_2_ loading process: (**a**) The overall preparation process of TiO_2_@CNF Aerogel, (**b**) Ti(OC_4_H_9_)_4_ was attracted by the polar groups of CNF Aerogel, and (**c**) TiO_2_ NPs loaded on the crosslinking part of CNF Aerogel.

**Figure 9 polymers-13-01841-f009:**
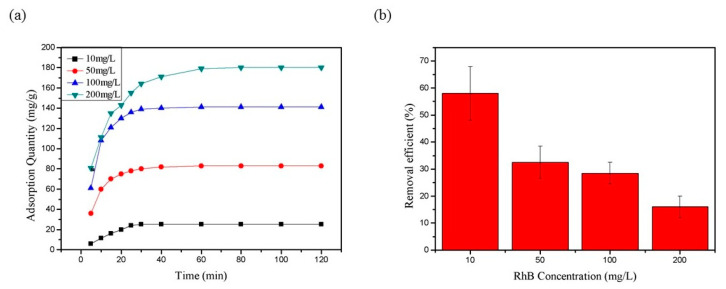
(**a**) Adsorption of RhB (10–200 mg/L) onto CNF Aerogel and (**b**) percentage of removal of RhB within 120 min.

**Figure 10 polymers-13-01841-f010:**
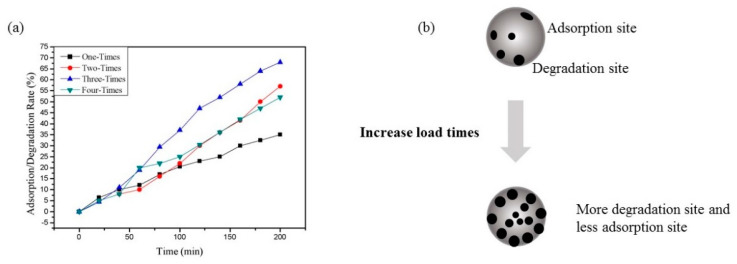
Influence of load times on photocatalytic removal of rhodamine B: (**a**) effect of TiO_2_ load times on the RhB (c = 200 mg/L) removal of the TiO_2_@CNF Aerogels, and (**b**) schematic diagram of the influence of different load times on pore structure.

**Figure 11 polymers-13-01841-f011:**
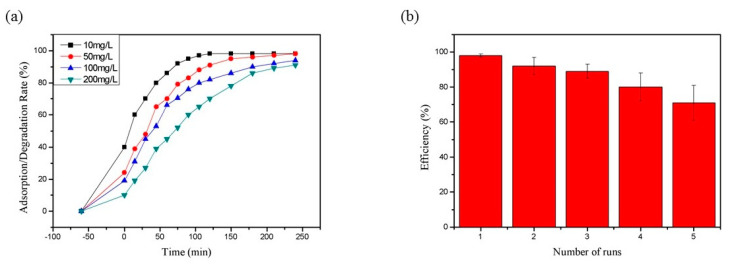
Performance of repetitive use of TiO_2_@CNF Aerogel: (**a**) Absorption and degradation of RhB by the TiO_2_@CNF aerogel, (**b**) The performance of aerogel recycling in 5 runs.

**Figure 12 polymers-13-01841-f012:**
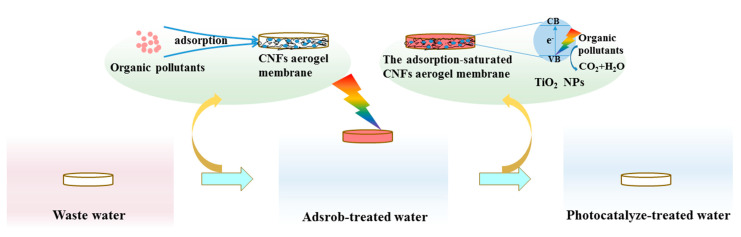
Schematic diagram of pollution treatment by TiO_2_@CNF Aerogel.

**Table 1 polymers-13-01841-t001:** Performance comparison of the TiO_2_@CNF Aerogel and cellulose materials in reference.

Composite Material	Adsorption Performance	Degradation Performance	Reference
Carboxycellulose nanofibers	★★★★		[4]
Carboxycellulose nanofibers	★★★★		[5]
ZnO/Microfibrillated Cellulose	★★★★		[8]
BiOBr/cellulose composite	★★	★★★★★	[21]
TiO_2_/CNC nanocomposites	★	★★★★★	[20]
TiO_2_@CNF Aerogel	★★★★	★★★★★	This work

## Data Availability

The data presented in this study are available on request from the corresponding author.

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
