# Peer review of "Construction of the Cellulose Nanofibers (CNFs) Aerogel Loading TiO2 NPs and Its Application in Disposal of Organic Pollutants"

_polymers, 2021, doi:10.3390/polym13111841_

Round 1

Reviewer 1 Report

Manuscript: Construction of the cellulose nanofibers (CNFs) aerogel loading TiO2 NPs and its application in disposal of organic pollutants

The manuscript describes a Excellent work AND is well presented. Authors need following points to be included before reconsideration.

1. Abstract should contain some quantitative information also.

2. English must be improved.

3. Novelty of the work be established.

  1.  All the important results reported be compared in a tabular form to establish the superiority of the work.

  1.  Authors must need to incorporate future prospective of the presented work in the conclusion part of the manuscript.
  2. Authors must need to incorporate some recent references in the introduction part of the manuscript related to the other applications of biopolymers to make it more interesting for the readers. For example.

  • Biomacromolecules2019, 20, 5, 2051–2057
  • Cellulose 25 (3), 1961-1973 (b)
  • ACS Sustainable Chemistry & Engineering 6 (3), 3279-3290
  • Industrial & Engineering Chemistry Research 56 (46), 13885-13893
  • Biomacromolecules 18 (8), 2333-2342
  • ACS Applied Materials & Interfaces 2020, 12 (19) , 22037-22049
  • Angewandte Chemie 2020, 9
  • DOI: 10.1021/bk-2020-1352.ch012
  • Advanced Sustainable Systems, 1900114
  • RSC advances 9 (69), 40565-40576

7. Authors need to include TGA studies for nanofibers (CNFs) aerogel.

Reviewer 2 Report

Authors in this paper, obtaining cellulose nanofibers aerogel through chemical crosslinking were explored. TiO2 nanoparticles were loaded on the cellulose aerogel through the sol-gel method combined with hydrothermal method.  The absorbed pollutants in this material were completely degraded by UV radiation.

The publication deals with two issues: obtaining TiO2 @ CNFs aerogel and the application of this material to decompose adsorbed RhB. While the first part is clear, the material has been characterized by a number of analytical techniques, and the results obtained confirm the structure of the material, the second part of the work on RhB absorption and degradation is not sufficiently documented.

Detailed questions about the publication:

  1. The synthesis of TiO2 @ CNFs aerogel material is very expensive and not ecological. What is the cost of producing 10/100/1000 kg of material because it is to be used on a large scale?
  2. What is the repeatability of the method of obtaining the material and its stability over time?
  3. What is the optimal material density for absorbing pollutants?
  4. The absorption and color decay of RhB on the obtained material under the influence of UV radiation were examined. What is the degradation product of RhB, what compounds are formed and should they be removed from the material before the next use of the material?
  5. Measurements of RhB degradation with TiO2 in solution under the influence of UV radiation were not performed as a reference.
  6. The conclusion that the impurities are decomposed on the obtained material is unfounded, the authors described only one compound inaccurately - RhB.
  7. How do the authors of the work plan to dispose of the material after application?
  8. What is the influence of this material - very photochemically active on the aquatic environment?

Round 2

Reviewer 2 Report

The authors introduced a number of changes to the text of the publication in line with the comments of the reviewer.
The paper it became acceptable after the authors addressing all the concerned issues.